# Study on the Solidification Behavior of Inconel617 Electron Beam Cladding NiCoCrAlY: Numerical and Experimental Simulation

**Jian Chen, Hailang Liu, Zhiguo Peng * and Jie Tang**

School of Electrical and Mechanical, Guilin University of Electronic Technology, Guilin 541004, China; 19012201004@mails.guet.edu.cn (J.C.); Liuhl@guet.edu.cn (H.L.); 20012302073@mails.guet.edu.cn (J.T.)
* Correspondence: pengzg@guet.edu.cn

**Abstract:** To better control the Inconel617 electron beam cladding solidification process, a three-dimensional temperature field model was built to simulate the temperature gradient, cooling rate, and solidification rate in the solidification process and take a deep dive into the solidification behavior, as well as the calculation of the solidification characteristic parameters at the edge of the molten pool and then predict the solidification tissue structure. The study shows that the largest temperature gradient occurred in the material thickness direction. The self-cooling effect of the material dominated the solidification of the alloy layer; the cooling rate depended on the high-temperature thermal conductivity of the material and the self-cooling effect of the matrix, and the maximum cooling rate in the bonding zone was 1380 °C/s. The steady-state solidification rate was equal to the moving speed of the heat source; the solidification characteristics of the solidification process at the edge of the molten pool increased with the distance from the surface: the cooling rate decreased from 1421.61 to 623 °C/s, the temperature gradient increased from $0.0723 \times 10^6$ to $0.417 \times 10^6$, and the solidification rate decreased from 0.01 to 0 m/s. The prediction was made that the small and thin equiaxed crystals are on the top, a thin and short dendritic transition structure in the middle, and relatively coarse dendrites at the bottom. Experiments confirmed that the solidification tissue structure is basically consistent with the simulation law.

**Keywords:** numerical simulation; electron beam cladding; process parameters; solidification characteristics; microstructure prediction

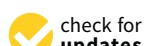

## 1. Introduction

Inconel617 is a nickel-based solid solution strengthened alloy with excellent, comprehensive high-temperature strength and oxidation resistance and corrosion resistance. It is mainly used in industrial and aviation steam turbine hot-end parts [1–3]. In the face of a complex working environment and product upgrade requirements, its oxidation and corrosion resistance under high-temperature conditions cannot fully meet the higher use requirements of the new generation of hot-end components [4,5]. Inconel617 alloy can meet the more stringent performance requirements as much as possible by electron beam cladding material surface modification and repair engineering. However, electron beam cladding is a non-equilibrium solidification phenomenon of rapid melting, and the dynamic characteristics and phenomena of the solidification process are difficult to express by means of experiments [6,7]. At the same time, the volume of the molten pool is a small proportion of the total materials, and the fluid temperature is much higher than the liquidus temperature. The solidification process is accompanied by complex physical and chemical changes, such as fluid convection, heat and mass transfer, grain precipitation, and growth. This complex phenomenon mainly stems from the thermal cycling characteristics in the molten pool in electron beam cladding. Therefore, the adoption of a temperature field to

study the solidification behavior of electron beam cladding is an indispensable direction in its process research.

The solidification behavior of electron beam cladding is a process in which the molten pool tends to be solid and the crystal structure precipitates and grows. Studying the temperature distribution and solidification characteristics of the solidification process is conducive to better controlling the process behavior. At the same time, according to the principle of rapid solidification, the solid–liquid interface temperature gradient, cooling rate, and solidification rate at the edge of the molten pool directly affect the solidification and forming of tissue. The morphology of the solidification structure is controlled by the ratio between the temperature gradient and the solidification rate, and the growth speed of the structure is controlled by the cooling rate. Due to the different characteristics of temperature changes in different areas of the cladding process, the organization shows different structures, which leads to changes in material properties.

The finite element simulation of the temperature field is adopted to research the thermal cycle of electron beam cladding at home and abroad mainly. Gong et al. [8] used Monte Carlo and a temperature field simulation model to reveal the electronic behavior and temperature distribution of the TiN coating on the surface of the bearing steel irradiated by an electron beam. It was shown that the temperature distribution pattern can be controlled by controlling the electron beam energy to improve the film base binding force. Tang [9] used ANSYS software to simulate the temperature field characteristics of a titanium alloy electron beam cladding silicide coating, established the relationship curve between the cladding process parameters and the temperature field, and determined the correlation between the cross-sectional penetration of the coating and the heat input. Riqing et al. [10] used finite element analysis to simulate the thermal phenomenon of the laser forming numerical model and used ABAQUS finite element analysis software to simulate the temperature field. The results showed that the temperature gradient distribution near the melt pool is regular, namely the temperature gradient decreases as the distance from the melt pool increases. Hailang et al. [11] used ANSYS software to simulate the temperature field of $NbSi_2$ cladding on the surface of nickel-based alloy 617 found that the process parameters have a great influence on the rapidly solidified coating, as well as explained the univariate law of temperature and process parameters combining experiment analysis. The above-mentioned researchers mainly focused on the influence of single process parameters on the distribution of the temperature field but did not investigate the characteristics of the solidification behavior of the electron beam melting process and the influence law of the solidification characteristics. Although Gaumann et al. [12] and Liu et al. [13] used the amount of solute at the time of solidification to estimate the temperature gradient and solidification rate, it is difficult to detect the solute distribution in the molten pool through experiments. Therefore, it is particularly important to use the temperature field model to study the solidification characteristic distribution of the cladding material and the change law of the solid–liquid interface solidification parameters so as to calculate the structure after solidification and to provide a certain theoretical basis for the electron beam cladding solidification behavior control.

## 2. Materials and Methods

### 2.1. Finite Element Model

APDL was used to establish the numerical model of NiCoCrAlY by electron beam melting on the Incone617 surface. According to the heat conduction characteristics of electron beam melting, the temperature field distribution shows symmetry when the material symmetry surface coincides with the beam scanning path, so the established finite element simulation model was half the size of the actual model. As shown in Figure 1a, the upper area of the model was the functional coating NiCoCrAlY, which had a size of 50 mm × 20 mm × 1 mm, while the lower substrate area was Inconel 617 alloy, which had a size of 50 mm × 20 mm × 10 mm. To consider both the computational efficiency and the research focus, a gradual mesh division was used. Figure 1b shows the outer surface unit

body on the model of Figure 1a (except the symmetry surface), which had a SOILD70 3D thermal effect cell in the interior and a SURF152 surface thermal effect cell with extra nodes in space on the surface. Finally, the matrix and coating model in the *x*-axis direction was divided into 100 layers equally; the matrix and coating model in the *y*-axis direction was divided into 10 layers with a spatial ratio of 14; the matrix model in the *z*-axis direction was divided into 10 layers with a spatial ratio of 13; and the coating model in the *z*-axis direction was divided into 5 equal layers.

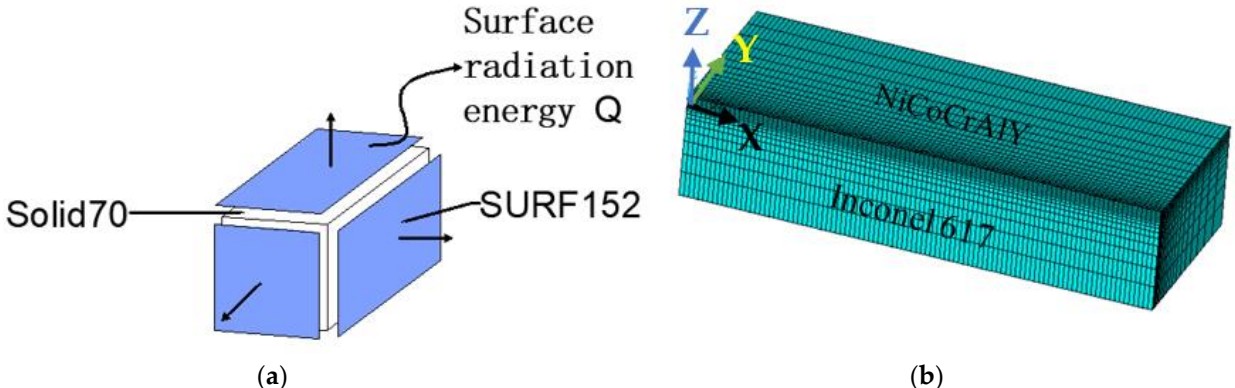

(a)                                                        (b)

**Figure 1.** Finite element model: (**a**) unit-type selection diagram and (**b**) grid model.

The material thermal property parameters change with temperature and have a large impact on the simulation of the surface molding process. The model thermal property parameters were mainly derived from relevant papers, while the parameters that were not accessible were determined with the help of JMatPro7.0 simulation calculations as well as interpolation and extrapolation methods The thermal properties of the Inconel617 substrate [14,15] are shown in Table 1. The density was 7320 kg/m³, and the melting point was 1305 °C. The thermophysical parameters [16,17] of NiCoCrAlY cladding material are shown in Table 2. The density was 7320 kg/m³, and the melting point was 1050 °C.

**Table 1.** Thermal properties of Inconel617 substrate.

| Temperature T (°C) | Specific Heat C (kg·°C) | Conductivity W (m²·°C) |
|---|---|---|
| 25 | 420 | 13.4 |
| 200 | 465 | 16.3 |
| 400 | 515 | 19.3 |
| 600 | 565 | 22.5 |
| 800 | 615 | 25.5 |
| 1000 | 665 | 28.7 |

**Table 2.** Thermophysical parameters of NiCoCrAlY cladding material.

| Temperature T (°C) | Specific Heat C (kg·°C) | Conductivity W (m²·°C) |
|---|---|---|
| 25 | 501 | 4.3 |
| 400 | 592 | 6.4 |
| 800 | 781 | 10.2 |
| 1000 | 764 | 17.6 |

*2.2. Initial Conditions and Boundary Conditions*

In this paper, the necessary assumptions were made in establishing the numerical calculation model. Specifically, these included [18]:

(1) The high-temperature gasification of the material and the flow of the molten pool are ignored.

(2) The electron beam energy deposition presents as a Gaussian distribution and only considers the heat radiation exothermic, ignoring the convective heat exchange with the air.

(3) The thickness and composition of the functional coating are evenly distributed, and the thermophysical parameters of the experimental materials change continuously with the temperature.

When electron beam cladding was performed, the initial temperature of the material was 25 °C and the ambient temperature of the vacuum chamber was also 25 °C. Since in the vacuum state, there is no convective heat transfer from the material surface to the environment and only surface radiation dissipates heat, the convective heat transfer inside the material was dominant and the thermal radiation was weak, so only convective heat transfer was considered for the internal conduction of the material. The heat source of electron beam melting adopted the Gaussian surface heat source.

The boundary conditions on the upper surface of the computational model were set, as shown in Equation (1). The electron beam processing heat source, as shown in Equation (2), was set on the surface of the three-dimensional solid element SOLID70. The radiation conditions shown in Equation (3) were applied to the surface effect cell Surf152 of SOLID70.

$$-\frac{\partial_T}{\partial_Z} = -Q(x, y, t) + h_r(T_w - T_e) \tag{1}$$

where k is the thermal conductivity of the material, $Q(x, y, t)$ is the moving Gaussian surface heat source function, $T_w$ is the surface temperature of the upper surface of the sample, $T_e$ is the ambient temperature in the vacuum chamber, and $h_r$ is the heat radiation coefficient.

$$Q(x, y, t) = \frac{3\eta UI}{\pi R^2} \exp\left(-\frac{3r^2}{R^2}\right) \tag{2}$$

where η is the thermal efficiency coefficient of electron beam processing, taking a value of 0.75 to 0.95; U is the electron acceleration voltage (KV) of the electron beam processing equipment; I is the beam current (mA); r is the polar diameter from the heating node to the center of the heat source in polar coordinate form (m); $r = \sqrt{(x-a)^2 + (y-b)^2}$ in rectangular coordinate form, where $a = v\{TIME\}$, $b = 0$; and R is the electron beam radius (m).

$$h_r = \sigma\delta\left(T_w^2 + T_e^2\right)(T_w + T_e) \tag{3}$$

where σ is the Stefan–Boltzmann constant, which is $5.67 \times 10^{-8}$ in the metric system, and Surf152 is the thermal emissivity of the sample surface. The surface effect unit needed to set the real radiation constant 1, and the shape factor was 1. Units were generated based on nodes, and K4 was set to a surface cell that did not contain intermediate nodes. K5 was the unit set to contain the external heat radiation node. K9 was the unit thermal radiation angle coefficient.

The XOZ plane is the symmetry plane of the solid model, which is mainly determined by the characteristics of the electron beam cladding. The temperature value of the symmetry plane is higher compared to the temperature on both sides, which belongs to the inside of the material. Therefore, the XOZ surface does not consider the effects of heat convection and heat radiation and is an ideal adiabatic surface. Except for the upper surface and the symmetry plane, only the exothermic effect of thermal radiation was considered on the other sample surfaces.

*2.3. Theoretical Basis of Temperature Field Simulation*
2.3.1. Energy Control Equation

The temperature field of Inconel617 electron beam cladding NiCoCrAlY was to simulate the fast melting and solidification process, and the processing process changed with time. However, each node in the unit body had a temperature degree of freedom that varied with time, and the overall model was a numerical model composed of all nodes,

thus forming the temperature field variable T = T(x,y,z,t); this research is mainly to establish a three-dimensional transient temperature field, choose any one finite element body in the discrete model, and establish the thermal conduction differential equation of the electron beam cladding temperature field according to the Fourier formula and the law of conservation of energy, as shown in Equation (4).

$$\frac{\partial}{\partial_x}\left(k_x\frac{\partial_T}{\partial_x}\right) + \frac{\partial}{\partial_y}\left(k_y\frac{\partial_T}{\partial_y}\right) + \frac{\partial}{\partial_Z}\left(k_z\frac{\partial_T}{\partial_Z}\right) + \frac{\partial_{Q_V}}{\partial_t} = \rho C\frac{\partial_T}{\partial_t} \tag{4}$$

where $k_x$, $k_y$, and $k_z$ indicate the conductivity of the solid material along each axis; T represents the temperature degree of freedom value $T(x, y, z, t)$ changing with time (°C); $\rho$ indicates the density of the solid material (kg); C represents the specific heat capacity of the material (kg·°C); and $Q_v$ indicates the internal heat generation rate of the entity, which mainly includes the heat generated by the external electron beam cladding heat source and the internal phase change latent heat of the molten pool $(J/(m^3 \cdot t))$. The radiation energy loss was calculated as shown in Equation (3) according to the Stephen–Boltzmann law.

### 2.3.2. Latent Heat of Phase Change

The electron beam cladding process is inevitably accompanied by the absorption and release of heat from the phase change process, and handling phase change analysis is mainly based on thermal compensation. Commonly used methods are the temperature rise method, the equivalent specific heat method, and the enthalpy method. The enthalpy method was adopted to save the amount of experiment and storage space. The enthalpy method refers to the integral of density multiplied by the specific heat capacity with respect to temperature, as shown in Equation (5). Then, the enthalpy was compensated to Equation (4) in $Q_v$.

$$H = \int_{T1}^{T2} \rho C(T) dT \tag{5}$$

where H is the enthalpy, C is the density, $\rho$ is the specific heat capacity function changing with temperature, T is the temperature function changing with time t and position, and $T_1$ and $T_2$ are the interval temperature values.

### 2.3.3. Temperature Gradient

The rate of change of temperature with spatial displacement is the temperature gradient, and the geometric meaning is shown in Figure 2. It is a one-dimensional vector value, and the positive and negative energy directly reflect the direction of the heat flow. Its magnitude not only affects the thermal cycle in the material but also has an important effect on the thermal stress of the solid material. The numerical expression of the temperature gradient is shown in Equation (6).

$$G = \frac{\Delta T_{(x)}}{\Delta x} = -\frac{\partial_T}{\partial_x} \tag{6}$$

In the formula, T(x) is the temperature distribution function in space, x is the distance from the center of the heat source on the experimental material, and the unit of the temperature gradient is °C/m.

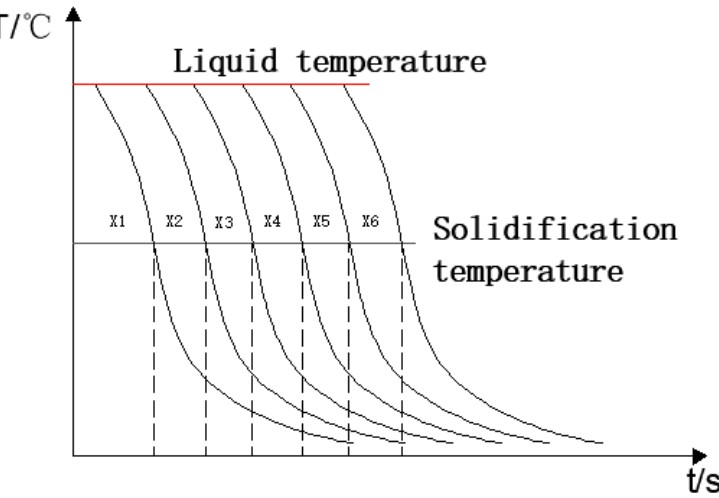

**Figure 2.** Temperature dynamic curve during cooling.

### 2.3.4. Cooling Rate

In the analysis of the transient temperature field of electron beam cladding, the temperature is distributed as a function of time history. The temperature change rate in the heating process is the heating rate, and the temperature change rate in the cooling process is the cooling rate. In other words, the cooling rate is the temperature lowered by the hot material per unit time. Equation (7) is the mathematical expression of the cooling rate.

$$\varepsilon = \frac{\Delta T}{\Delta t} = \frac{\partial_T}{\partial_t} \tag{7}$$

where T represents the temperature function of a certain node in the electron beam cladding process and t represents the time history function of the cladding process.

### 2.3.5. Solidification Rate

The solidification rate refers to the speed of the solid–liquid interface during the solidification process when the molten material is heated and melted. As shown in Figure 2 for the cooling process temperature dynamic curve, according to the cooling process temperature-time curve, the solid line temperature of the solidification point (the melting point of the material) and the curve intersection were marked in the (x,t) coordinate system to obtain the solidification dynamic curve. The change rate of the solidification dynamic curve is the solidification rate, that is, the advancing speed of the solidification interface per unit time. Therefore, the mathematical expression of the advancing speed of the solidified interface at the tail of the electron beam cladding molten pool is shown in Equation (8). That is the maximum solidification rate of the electron beam cladding process.

$$v = \frac{x}{t} \tag{8}$$

where x represents the moving displacement of the molten pool interface and t represents the time history.

### 2.4. Electron Beam Cladding Experiment

The substrate Inconel617 was wire-cut to 50 mm × 40 mm × 10 mm, grinding to eliminate cutting marks. To avoid the surface powder flying caused by bombardment of the specimen with high-energy electron beam particles, the NiCoCrAlY alloy powder was preset by supersonic plasma spraying. First, the matrix material was sandblasted before presetting and then put in an ethanol solution to clean surface impurities. Second, a modified layer of NiCoCrAlY powder of 50 mm × 40 mm × 1 mm was preset on the substrate by plasma spraying. Finally, the high-energy electron beam equipment

(SEB(J)6/60/40/30, Shichuang Vacuum Equipment Co., Ltd., Guilin, China) of the Guilin University of Electronic Science and Technology was used to melt NiCoCrAlY to the substrate microfusion state, thereby forming a metallurgically bonded surface modification layer. The vacuum degree of the cladding operation chamber was $3.2 \times 10^{-2}$ Pa, and the vacuum degree of the gun chamber was $1.6 \times 10^{-3}$ Pa.

First, the electron beam cladding test block was wire-cut to 1 cm × 1 cm in size, and the cut section was polished by sandpaper and processed by a metallographic polishing instrument (Zhengzhou Zhuotai Testing Equipment Co., Ltd., Zhengzhou, China) until the mirror effect appeared. Then, the test block was washed in clean water, wiped with alcohol, and dried with a hair dryer. The test block was prepared for scanning electron microscope and Energy Dispersive Spectrometer observation and Vickers hardness test.

A Quanta 450 FEG field emission scanning electron microscope (FEI Company, Hillsboro, OR, USA) was used to observe the microscopic morphology of the polished section before and after electron beam cladding. An EDS analyzer (FEI Company, Hillsboro, OR, USA) was used to characterize the distribution of elemental components in the section after cladding. The Vickers hardness of the sample section before and after electron beam cladding was measured by an hv-1000 microhardness tester (Haoxinda Instrument Co., Ltd., Shenzhen, China). The experimental method was to load 200 g of a diamond indenter, hold for 20 s, and then end. This method was followed to test along the thickness direction of the specimen.

## 3. Results and Analysis

### 3.1. Characteristics of Solidification Behavior of Electron Beam Cladding

The adopted process parameters of scanning power P = 1800 W, scanning speed v = 10 mm/s, and beam spot diameter D = 4 mm were used to study the solidification behavior characteristics of the cladding process, namely temperature gradient, cooling rate, and solidification rate.

The process state at processing time t = 2.5 s was taken to study the temperature gradient during the steady state of cladding. Figure 3 is a temperature gradient curve. In Figure 3a, three curves separately represent the temperature gradient in the *x*-axis direction, *y*-axis direction, and *z*-axis direction when the electron beam was scanned. At the point X = 0.025 m, the temperature gradient GX was close to 0, and the temperature gradients on both sides increased in opposite directions, but the gradient in front of the molten pool was much larger than the temperature gradient at the tail; the temperature gradient was approximately zero when the distance exceeded 0.03 m. The main reason is that the electron beam current has not yet been scanned here; the temperature gradient during 0~0.02 m tended to be flat. The main reason is that after electron beam scanning, there are two processes, heat conduction and thermal radiation heat transfer, so the temperature gradient gradually decreases; the temperature gradient GX varied in the range of $0$–$1.1 \times 10^6$ °C/m. The temperature gradient GY was the temperature spatial change rate on the line x = 0.025 of the cladding layer; the temperature gradient in the *y*-axis direction changed within the range $0$–$1.38 \times 10^6$. At the range of 0–0.001 m, the temperature gradient rose due to the Gaussian distribution of the electron beam heat source model. The temperature gradient near the heat source begins to decrease; when approaching the solid–liquid phase transition junction, the latent heat of phase change decreases after the heat release of the solidification process causes the temperature gradient fluctuation of the solidification process to increase, and gradually becomes flat. The temperature gradient GZ was the change in temperature of the path on the symmetry plane x = 0.025 m, which gradually stabilized as the distance from the surface increased. The difference in thermal properties of the two different materials caused the temperature gradient GZ to fluctuate, and the gradient value was in the range of $0$–$3.19 \times 10^6$. From the perspective of temperature gradients in different directions, the temperature gradient at the surface of the molten pool in the *z*-axis direction is the largest, that is, the heat flow along the depth direction is the largest, and the self-cooling effect of the material is the dominant element of cooling and solidification.

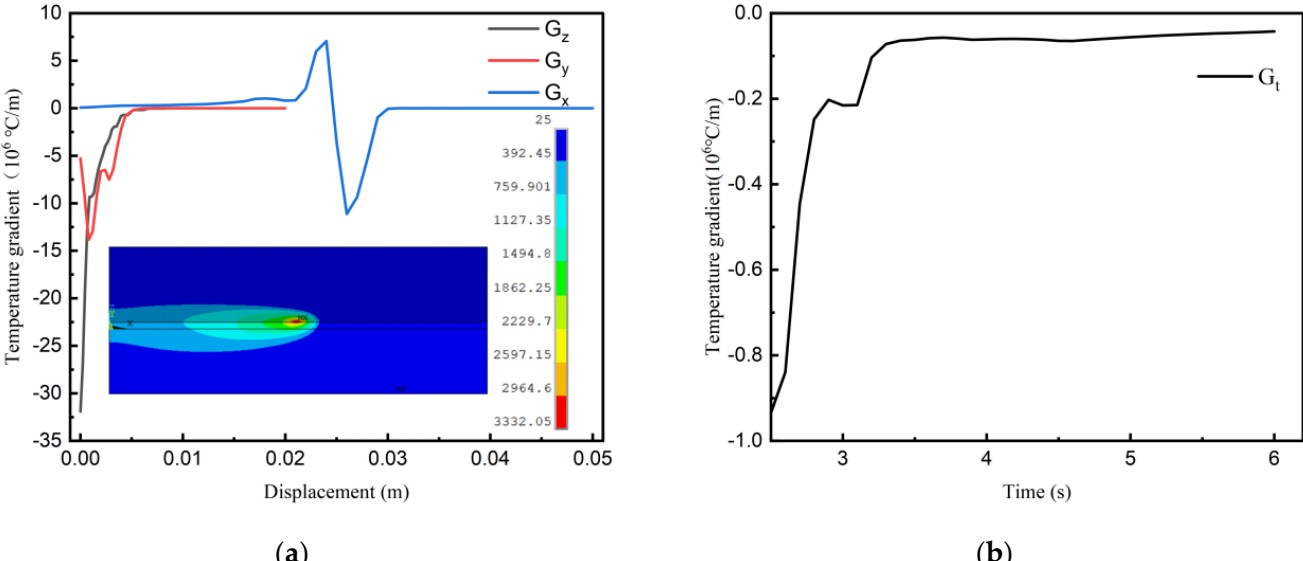

**Figure 3.** Temperature gradient curve: (**a**) temperature gradient curves in different directions and (**b**) transient temperature gradient.

Figure 3b shows the transient temperature gradient of a node on the scan path of the model. The node started to enter the cooling solidification process after 2.5 s, so the temperature gradient history change was taken from 2.5 s to 6.0 s. It can be observed from the figure that there was a short, steady process in the temperature gradient of 3.0~3.1 s. The main reason is that the latent heat exothermic process of the phase change of the material makes the temperature gradient change smaller. During 2.8~3.0 s in the molten pool liquid temperature cooling process, the temperature gradient decreased with time. After 3.1 s in a quasi-solid state, the temperature gradient gradually decreased and tended to 0 with time.

The cooling rate is the rate of change of temperature with time, which directly affects the crystal growth and the release of thermal stress during the solidification of the metal, affecting the physical and chemical properties of the material. Therefore, it is particularly important to study the cooling rate change law of the Inconl617 electron beam cladding NiCoCrAlY functional coating. As shown in Figure 4a,b, the cooling temperature and rate data came from the junction of the simulation model matrix and the coating material, and the positions are the nodes at 0.002, 0.025, and 0.048 m in the scanning direction. As shown in Figure 4a, the temperature–time history curve showed that the temperature during the electron beam cladding solidification process decreases rapidly with time and enters a slow cooling process when the temperature reaches about 200 °C. Research and analysis found that the cooling process in 0–20 s is mainly influenced by the heat conduction inside the material and the radiation exotherm on the surface of the material; after 20 s, the temperature gradient of the entire sample gradually tended to be uniform, and the heat conduction effect of the material was small, which mainly depends on heat radiation on the surface of the material. Figure 4b shows the cooling rate curve corresponding to the time history temperature curve of the solidification process. The node cooling rate at x = 0.002 m first increased and then gradually decreased. The time history cooling curve of the node at x = 0.025 m and x = 0.048 m had two oscillations, which is one more rate fluctuation compared to the node at x = 0.002. The higher temperature at the front of the molten pool during the electron beam heating process causes the thermal conductivity to be higher than the initial processing. In addition, the maximum cooling rate of the three nodes at different positions increases, which is mainly caused by the gradual increase in the preheating temperature of the sample during the process, and the thermal conductivity also increases with the temperature. However, no matter how the cooling rate changes, as long as the material enters the quasi-solid state, the cooling rate enters a steady decline and

gradually tends to zero. The reasons summarized for the fluctuation of the cooling rate are these: (1) The thermophysical parameters of the material change with the temperature, (2) the release of the latent heat of the phase change of the two materials delays the cooling rate, and (3) the preheating of the material causes the thermal conductivity of the sample to increase.

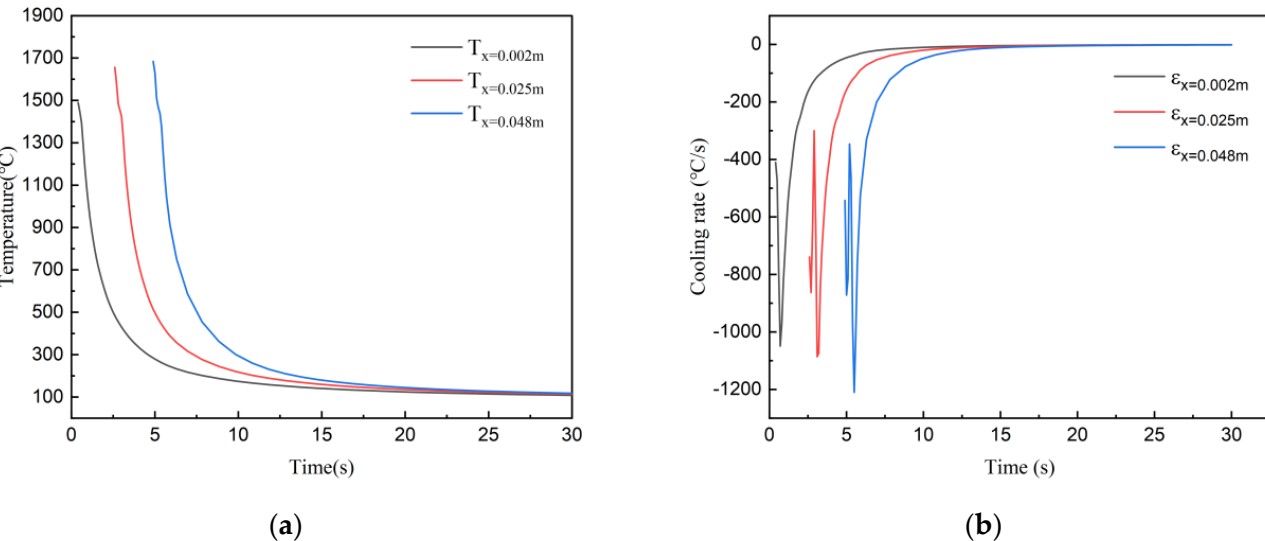

**Figure 4.** Cooling temperature and rate: (**a**) transient solidification temperature and (**b**) transient cooling rate.

What it takes to determine the solidification tissue structure and cladding forming properties is the size of the solidification rate. The solidification rate can actually be equivalent to the transition speed of the solid–liquid conversion interface. Previous research results have known that the molten pool is small. Therefore, the simulation mainly studied the data changes in x to the scanning path so as to avoid the solidification rate simulation error due to the small melting depth and melting width. Figure 5 shows the solidification rate curve. The overall trend in the solidification rate are mainly divided into three stages: In the first stage, the solidification rate is in the rising stage 2 s before, because the initial heating process of the electron beam is unstable, and the heat source is moving. The scanned area still has a great influence, and the material has a high thermal conductivity under high-temperature conditions, causing subsequent energy to continue to be conducted in the opposite direction of the scan. As the tailing phenomenon of the molten pool becomes larger, the solidification rate appears to rise. In the second stage, the solidification rate is basically stable, indicating that the electron beam cladding process has entered a steady state, the shape of the molten pool tends to be stable, and the solidification rate value is close to 0.01 m/s. In the third stage, the cladding solidification rate rises again. The reason is that the heat input of the electron beam cladding process is terminated at 5 s, and the entire sample is in no heat input, only heat diffusion and heat output.

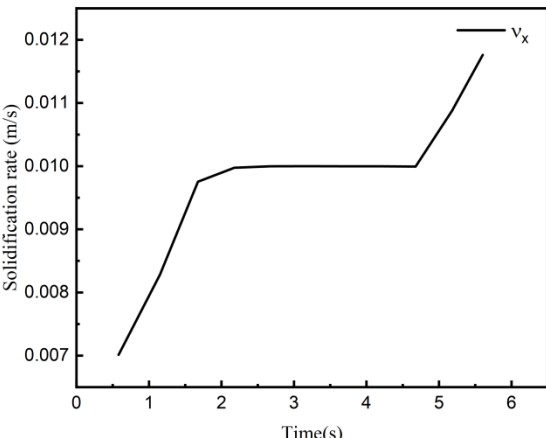

**Figure 5.** Solidification rate curve.

### 3.2. Electron Beam Cladding Solidification Behavior at the Edge of the Molten Pool

The direction of electron beam cladding solidification is affected by the direction of the heat flow, and the temperature gradient and cooling rate determine the speed of the heat flow. The solidification direction is opposite to the heat flow direction, and the heat flow direction is opposite to the temperature gradient direction, that is, the solidification direction is the same as the direction of the normal at the liquid–solid transition cross section. From the simulation, it can be seen that the electron beam cladding was "comet shaped" during the stabilization process, and the size of the molten pool size was constant. Therefore, the geometry of the solidification interface did not change along the scanning direction of the electron beam. The molten pool surface has a maximum solidification rate, the direction refers to the direction in which the heat source moves, and the size is equal to the scanning speed. The moving speed of the solidification interface inside the molten pool is the solidification speed S at the corresponding position. Figure 6 shows a schematic diagram of the solidification speed of the liquid–solid interface at the back of the molten pool. It was found that the solidification speed at the interface and the moving speed of the molten pool present a cosine function relationship, and the angle is the solidification direction angle. Therefore, the solidification speed at different positions of the solidification boundary from the surface of the molten pool and the moving speed of the molten pool show a relationship as shown in Equation (9). The solidification velocity vector angle from the surface to the bottom of the molten pool transitioned from 0° to 90°, so the solidification velocity S value decreased from 0.010 m/s to 0 m/s.

$$S = \upsilon \cos \theta \qquad (9)$$

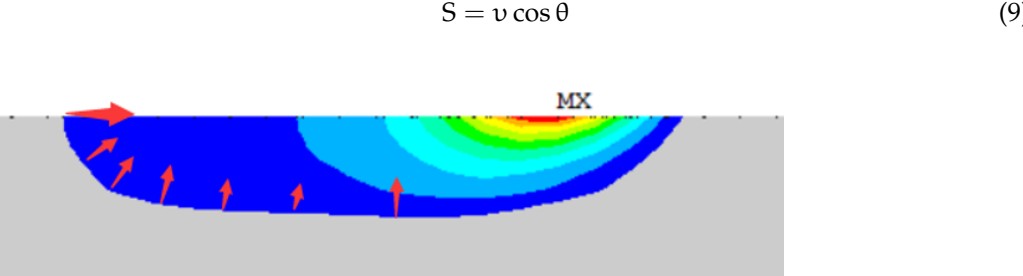

**Figure 6.** Solidification velocity direction of the liquid–solid interface.

Figure 7 shows the solidification characteristics of the solidification interface of the steady molten pool at different positions from the cladding surface, and the dual *y*-axis is used to represent the cooling rate $\varepsilon$ and the temperature gradient G at the boundary of the molten pool. It can be seen from the figure that the cooling rate curve showed a downward trend from the surface of the molten pool to the bottom, the surface cooling rate was the largest, the bottom cooling rate was the smallest, and its value dropped from 1421.61 to

623 °C/s. This phenomenon stems from the fact that the surface of the melt pool has a high temperature thermal radiation effect and is warmer than the interior, so the surface heat flow is significantly higher than the inside of the molten pool. Due to the characteristics of the energy distribution of the heat source and the thermophysical properties of the temperature change, it can be seen from the figure that the temperature gradient of the liquid–solid conversion interface increased with the increase in the distance from the molten pool surface and the temperature gradient value increased from $0.0723 \times 10^6$ to $0.417 \times 10^6$.

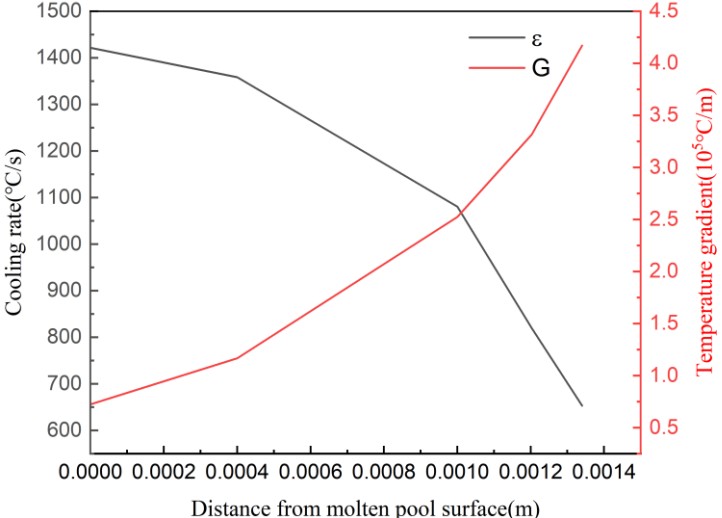

**Figure 7.** Solidification characteristics at different positions of the weld pool boundary.

Through the temperature field simulation, the characteristic parameters of the edge solidification of the NiCoCrAlY molten pool formed on the surface of the 617 nickel-based alloy were simulated. It was found that as the distance from the molten pool increases, the cooling rate decreases, the temperature gradient increases, and the solidification rate decreases. At the same time, analysis of the temperature gradient at the back of the molten pool found that the surface vector direction is consistent with the electron beam movement direction (*x*-axis), and the vector direction at the bottom of the molten pool is perpendicular to the surface (*z*-axis). According to the theory of dendrite growth, the dendrites near the metallurgical bonding zone grow along the *z*-axis direction, and the dendrites on the surface of the alloy layer grow along the *z*-axis direction. According to solidification theory, the characteristic parameters of solidification affect the structure and size distribution of the electron beam cladding layer. As shown in Figure 8, the solidification organization and size were related to the characteristic parameters, along the depth of the molten pool; as the temperature gradient G increased and the solidification velocity v decreased, the shape control factor describing the solidification structure gradually increased. At the same time, as the cooling rate in the depth direction decreased, the size of the structure changed from fine to thick. Based on this, it can be predicted that the molten pool surface k is small and $\varepsilon$ is large, causing the top of the molten pool to show mainly fine isometric crystals. The bottom of the molten pool has a large k and a small $\varepsilon$, resulting in a relatively coarse dendrite at the bottom; in the middle of the molten pool, k changes from small to large and $\varepsilon$ changes from large to small, so the middle part is mainly manifested as transitional short dendrites.

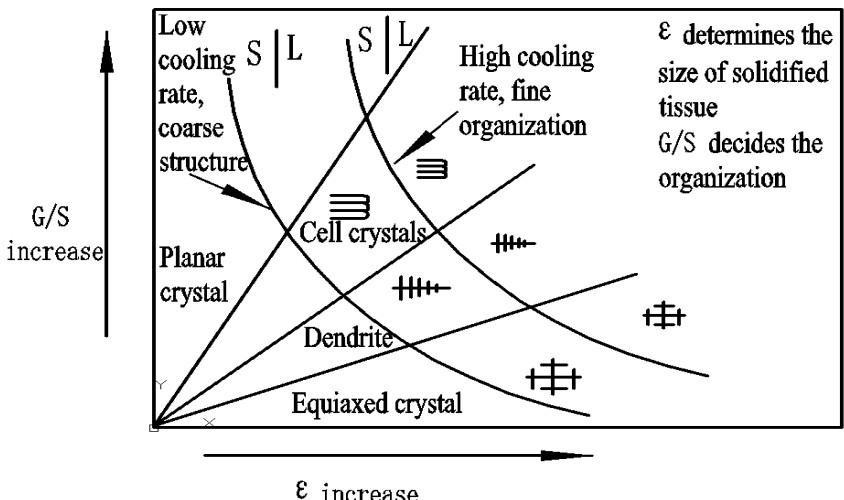

**Figure 8.** Solidification structure and the relationship between its size and characteristic parameters.

*3.3. Experimental Verification*

The electron beam cladding material was preprepared as the NiCoCrAlY composite coating to the substrate Inconel617 surface using supersonic plasma spraying. Figure 9a is the cross section before electron beam cladding. It is obvious that there were pores, cracks, and boundaries at the metal junction of the substrate and the prefabricated layer, as well as many defects in the prefabricated layer, which will cause poor material performance, cracks in actual use, and expansion and peeling of the coating. Figure 9b shows the cross-sectional morphology after electron beam melting, and the melt pool was free of obvious cracks, pores, and other defects. The substrate and the coating formed a dense metallurgical bonding layer, and the outline of the melt pool was clearly visible. Area 1 is the modified zone, and area 2 is the metallurgical bonding zone. Figure 9c shows the temperature distribution cloud of the melt pool interface obtained from the numerical simulation, and the melt pool was inside the white boundary, which had an elliptical shape due to the uneven heating temperature distribution, resulting in different thermal physical parameters within the melt pool. Comparing Figure 9b,c, the actual molten pool profile was more complicated than numerical calculations due to thermal deformation at the edge of the molten pool, artifact creep, and uneven electron beam energy distribution in the experiment. However, it was found that the size and shape of the melt pool in the simulation and experiment are basically the same region.

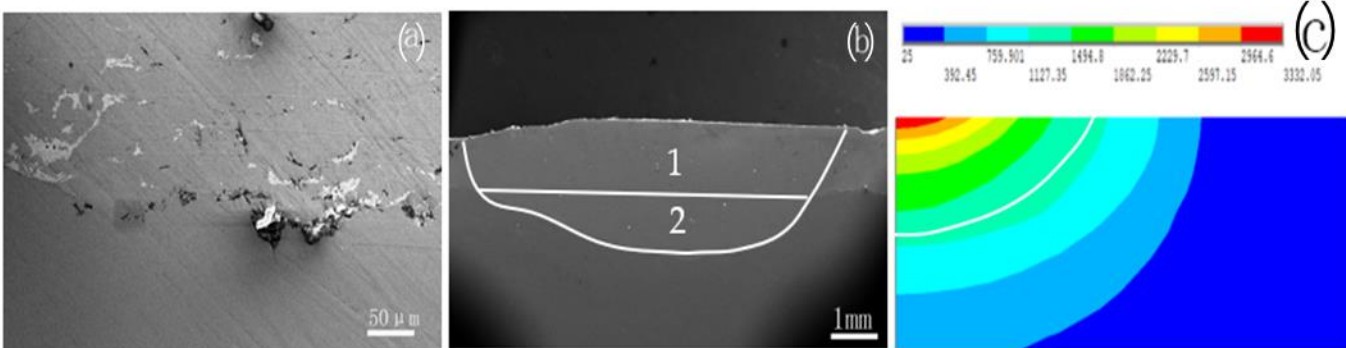

**Figure 9.** Before and after cladding and the cross section of the simulated molten pool. (**a**) Before cladding; (**b**) After cladding; (**c**) Cladding simulation.

Figure 10 shows the morphology of the molten pool. The left side shows the overall morphology of the molten pool without obvious cracks, but there were a small number of pores. The crystals showed an increasing trend from the top to the bottom, which is

basically consistent with the previous prediction trend in the size of the solidified structure. This phenomenon is due to the reduced cooling rate of the melt pool along the depth direction. Figure 10a shows that the top structure of the molten pool was mainly tiny equiaxed crystals and the downward structures were short and thin dendrites. Figure 10c shows that the structure at the bottom of the molten pool was relatively coarse dendrites. In addition, Figure 10b shows a thin and short dendrite transition structure in the middle of the molten pool. The distribution of the microstructure in the molten pool basically verifies the change trend of the solidification characteristic parameters of the molten pool and conforms to the distribution state of the microstructure predicted by solidification theory.

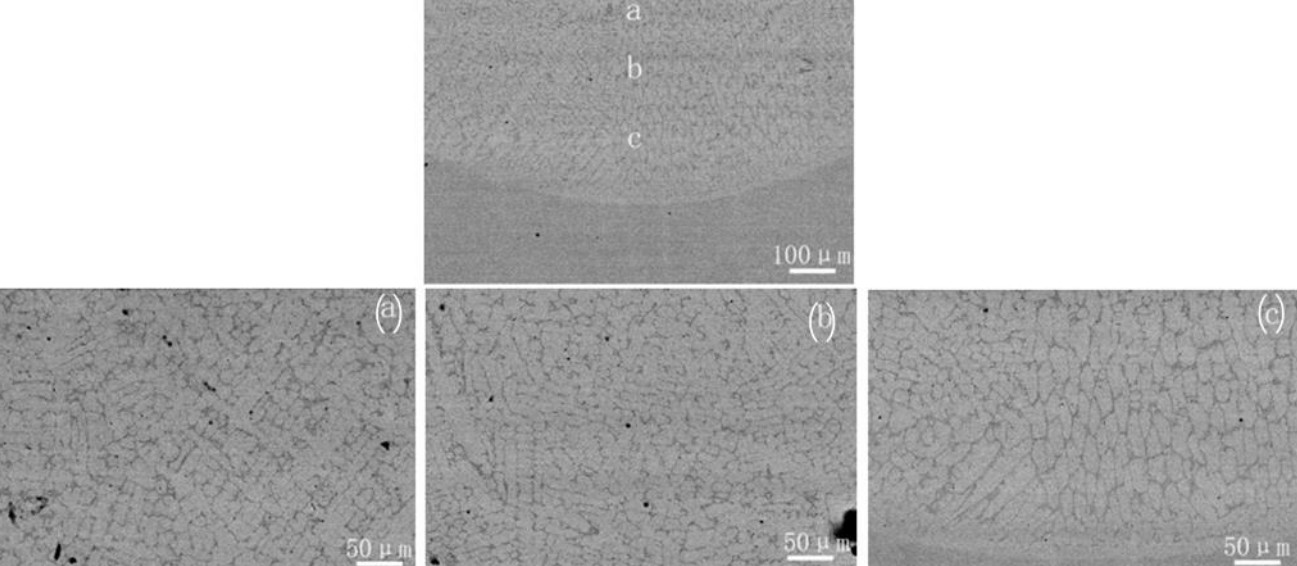

**Figure 10.** Molten pool structure morphology. (**a**) Top area of the molten pool; (**b**) Middle area of the molten pool; (**c**)The bottom area of the molten pool.

Figure 11a is the cross-sectional morphology of the weld pool bonding area. It is obvious that the coating had a good metallurgical bonding effect with the matrix. Compared with the structure of the bonding area before electron beam cladding (the name before cladding is written here), the structure of the bonding area after cladding was excessively flat. Figure 11b is EDS mapping spectrum diagram of the combined zone, and it can be viewed the distribution of elements both from the substrate and the surface coating. This is attributed to the mass transfer between the matrix and the upper coating during the molten process. The interdiffusion enable good metallurgical bonding at the interface, which is helpful to reduce the peeling possibility of the coating in harsh environment, further reducing the coating cracking and peeling possibility in a harsh environment.

A micro-hardness tester was used to measure the distribution of sectional hardness values before and after electron beam cladding. As shown in Figure 12, the two curves respectively represent the hardness of the coating preset by thermal spraying (before cladding) and the coating after electron beam cladding in the thickness direction. Among them, the hardness curve of the thermal spraying preset coating shows that the hardness value in the range of 0~0.8 mm was 470~530 $Hv_{0.2}$ and the hardness value changed gently; the hardness in the range of 0.8~1.2 mm dropped sharply to 230 $Hv_{0.2}$, and the hardness in other areas far from the preset coating was 230 $Hv_{0.2}$.

However, after electron beam cladding, the hardness values in the 0~0.8 mm range were 605~760 $Hv_{0.2}$, and the hardness values of the modified zone decreased gradually along the thickness direction. The hardness values in the 0.8~1.6 mm metallurgical bonding range were 490~610 $Hv_{0.2}$, and the hardness values decreased steadily. The hardness values of the heat-affected zone and the matrix in the range of 1.6~3.0 mm decreased slowly to 230 $Hv_{0.2}$. Compared with the data in the figure, it was found that the hardness of the

coating preset by electron beam cladding is higher than that by thermal spraying, and the hardness of the transition region decreases slowly. It was found that the metallurgical bonding of electron beam cladding is good, and the microstructure of the transition zone leads to a slow decrease of the microhardness of the specimen after cladding. The change in hardness is caused by the transition from fine grains on the surface to coarse grains in the bonding zone. Due to the rapid melting and solidification of electron beam treatment and metallurgical combination, the hardness of the modified zone increased by about 200 $Hv_{0.2}$ after electron beam treatment compared with that before.

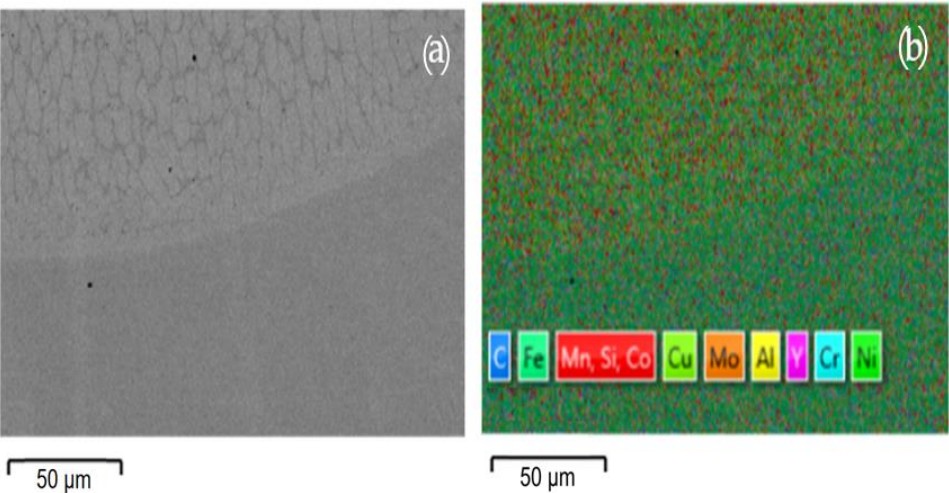

**Figure 11.** EDS of the clad bonding zone: (**a**) bonding zone morphology and (**b**) bonding zone energy spectrum.

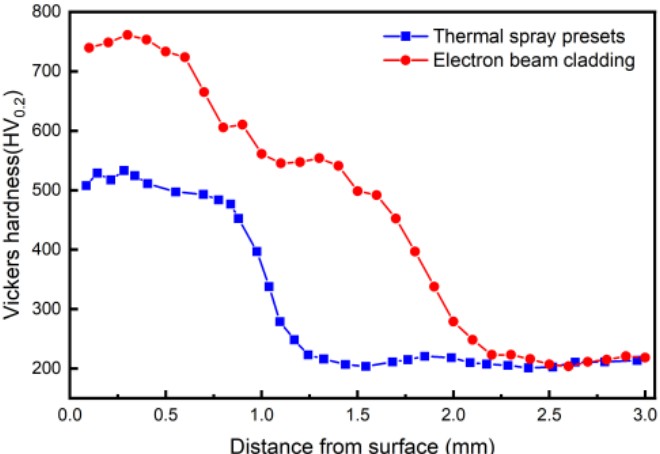

**Figure 12.** Section depth direction hardness.

## 4. Conclusions

(1) The temperature gradient $G_X$ of the molten pool is close to 0 at the center of the scan, and the temperature on both sides first increases and then gradually decreases. The influence of the temperature gradient $G_Y$ first increases and then decreases as it moves away from the center of the molten pool, and the boundary of the molten pool fluctuates due to the latent heat of phase change. The temperature gradient $G_Z$ decreases as it moves away from the center of the molten pool. From the perspective of temperature gradients in different directions, the maximum temperature gradient at the surface of the molten pool in the z-axis direction is $3.19 \times 10^6$ °C/m, that is, the heat flow along the depth direction is the largest, and the self-cooling of the material dominates the solidification of the alloy layer.

(2) The maximum cooling rate is higher at the backward point due to the accumulation of energy in the scanning direction. The cooling rate at different points in the molten pool has two numerical oscillations due to preheating and phase change exotherm, and the cooling rate gradually decreases when it enters the quasi-solid state.

(3) The maximum solidification rate gradually increases to 0.01 m/s with time, and then enters the stable stage. As the end of the process approaches, the solidification rate increases further to 0.012 m/s.

(4) The cooling rate at the edge of the molten pool decreases from 1421 to 623 °C/s, the temperature gradient increases from $0.0723 \times 10^6$ to $0.417 \times 10^6$, and the solidification velocity decreases from 0.01 to 0 m/s as the distance from the pool varies. According to the prediction of solidification theory, the top of the molten pool mainly presents fine and small equiaxed crystals, the bottom presents relatively coarse dendrites, the middle presents fine and short dendritic transition structures, and the crystals show a growing trend from the top to the bottom. The predicted organization structure is basically consistent with the experimental results.

(5) Vickers hardness in the thickness direction of the specimen shows a decreasing trend after electron beam cladding, from 756 $Hv_{0.2}$ on the surface to 230 $Hv_{0.2}$ on the substrate.

**Author Contributions:** J.C. was responsible for experiments and data processing and writing and revising papers. H.L. guided the revision of the thesis and also recommended and provided funding projects. Z.P. helped solve experimental problems and provided project and fund support. J.T. assisted in translation, formal analysis, visualization, and review. All authors have read and agreed to the published version of the manuscript.

**Funding:** This work was funded by the National Natural Science Foundation of China (NSFC; No. 51965012).

**Institutional Review Board Statement:** Not applicable.

**Informed Consent Statement:** Not applicable.

**Data Availability Statement:** No new data were created or analyzed in this study. Data sharing is not applicable to this article.

**Conflicts of Interest:** The authors declare no conflict of interest.

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
