# Peer review of "Study on the Solidification Behavior of Inconel617 Electron Beam Cladding NiCoCrAlY: Numerical and Experimental Simulation"

_coatings, doi:10.3390/coatings12010058_

Round 1

Reviewer 1 Report

The advantage of the work is the comparison of the theoretical diagram obtained in the work (Fig. 8) and the morphology of solidification (Fig.10). However, there are many inaccuracies.

Remarks

  1. When setting the problem, there are many inaccuracies because of this, it is difficult to read the article: there is
  • no coordinate system,
  • the dimensions of heat capacity and thermal conductivity are strange,
  • the letter ε was used in equations (3) and (7) in different ways,
  • it is not clear that solve the equation (4) or equation with enthalpy
  • the source is moving along the x-axis or y-axis,
  • meaning K4, K5, K9 (line 164, 165) is not clear,
  • how is it photo in figures 9 and 10?
  1. In Fig. 9b, two borders and three zones are clearly visible, to which they correspond, while the contours of the border are more complex than ellipses.
  2. Fig. 9c shows the calculated isotherms close to ellipses.
  3. The authors declare the adequacy of the experiment and calculation in what sense?

Author Response

Dear Editors and Reviewers:

    Thank you for your letter as well as the reviewers’ comments concerning our manuscript entitled “Study on the Solidification Behavior of Inconel617 Electron Beam Cladding NiCoCrAlY” (ID: 1522840). Those comments are all valuable and very helpful for revising and improving our paper, as well as the important guiding significance to our researches. We have studied comments carefully and have made correction which we hope meet with approval. Revised portion are marked in red in the paper. The main corrections in the paper and the responds to the reviewer’s comments are as flowing:

Responds to the reviewer’s comments:

1、Response to comment:

When setting the problem, there are many inaccuracies because of this, it is difficult to read the article: there is no coordinate system,the dimensions of heat capacity and thermal conductivity are strange,the letter ε was used in equations (3) and (7) in different ways,it is not clear that solve the equation (4) or equation with enthalpy the source is moving along the x-axis or y-axis,meaning K4, K5, K9 (line 164, 165) is not clear,how is it photo in figures 9 and 10?

Response:

Thank you very much for your valuable corrections, which I have tried my best to make. The coordinate system is added in Figure 1(b) in line 116.

Regarding the numerical problem of heat capacity and thermal conductivity, I checked the relevant content of the paper references [16-17] again, and found that there was indeed a writing error in the previous citation process, so I made it in line 130 of the article Modified. 

Equations (3) and (7) have the same sign, which I will change in line 165.

Equation (4) and equation (7) are explained in line 196 and added in line 207.

The heat source moving direction is my writing error, I will change it in line 163 to move along the X direction.

k4, k5, k9 are cell property settings, change and explain in line 169~171.

How is it photo in figures 9 and 10?

2、Response to comment:In Fig. 9b, two borders and three zones are clearly visible, to which they correspond, while the contours of the border are more complex than ellipses.

Response: Thank you for your detailed suggestion, we have modified Fig. 9b and explained in line 497~499.

3、Response to comment: 9c shows the calculated isotherms close to ellipses.

Response: The shape of the ellipse presented in Fig. 9c, we explained and illustrated in lines 494~497 , respectively.

4、Response to comment:The authors declare the adequacy of the experiment and calculation in what sense?

Response: Taking your suggestions into account, we discussed the study in depth and found some problems. Therefore, we have added the analysis of the energy spectrum(EDS) of the bonding area from line 537 to line 547,the analysis of the hardness of the cross-section before and after the melting, and the study of the distribution of the hardness of the cross-section after the melting from line 551 to line 572.

Reviewer 2 Report

The authors reported the Study on the Solidification Behavior of Inconel617 Electron 2 Beam Cladding NiCoCrAlY. The manuscript is well prepared and presented. However, needs the following revisions:

  1. Revise the title "Study on the Solidification Behavior of Inconel617 Electron 2 Beam Cladding NiCoCrAlY: Numerical and Experimental.
  2. What about the mechanical properties of cladded layer.
  3. Thermal stress analysis is missing in the FEA modeling and simulation.
  4. What is the cladded layer thickness in experimental research? Compare the same with numerical analysis.
  5. Detailed experimental investigation is required for better understanding. So, author must include the detailed characterization (Microstructure, EDS, XRD, layer thickness, etc)
  6. Mechanical properties are not reported for cladded thickness.
  7. I strongly advise authors to provide ethe detailed study of experimnetal and simulation part.

Author Response

Dear Editors and Reviewers:

    Thank you for your letter as well as the reviewers’ comments concerning our manuscript entitled “Study on the Solidification Behavior of Inconel617 Electron Beam Cladding NiCoCrAlY” (ID: 1522840). Those comments are all valuable and very helpful for revising and improving our paper, as well as the important guiding significance to our researches. We have studied comments carefully and have made correction which we hope meet with approval. Revised portion are marked in red in the paper. The main corrections in the paper and the responds to the reviewer’s comments are as flowing:

Responds to the reviewer’s comments:

1、Response to comment: Revise the title "Study on the Solidification Behavior of Inconel617 Electron 2 Beam Cladding NiCoCrAlY: Numerical and Experimental.

Response: Based on your suggestion, we have made a change in the title of line 3.

2、Response to comment:What about the mechanical properties of cladded layer. 

Response:Thank you for your valuable comments, we have identified the shortcomings of the paper. Therefore, the hardness experiments of the modified specimen cross-section have been added in lines 570~572.

3、Response to comment: Thermal stress analysis is missing in the FEA modeling and simulation.

Response:For the aspect of thermal stress analysis, this thesis does not cover thermal stress study at present, however, thank you for your reminder that we will conduct relevant simulations and experiments in our future research. Thank you for your useful suggestions for our subsequent research work.

4、Response to comment:What is the cladded layer thickness in experimental research? Compare the same with numerical analysis.

Response:In comparison with the numerical analysis, SEM was mainly used in the experiments to verify the solidification tissue morphology predictions given in the numerical analysis section(line 507), while hardness experiments in the coating thickness direction were added, and the Vickers hardness decreased depending on the tissue morphology in the section thickness direction (line 551).

5、Response to comment:Detailed experimental investigation is required for better understanding. So, author must include the detailed characterization (Microstructure, EDS, XRD, layer thickness, etc)

Response:We added EDS surface energy spectroscopy of the bonded area of the molten layer in line537~547 after studying your valuable suggestions in depth. Figure 9 and Figure 10 analyze the organization before and after modification. At the same time, the comparative analysis of the hardness change before and after the cladding is added on this basis in line 551~572.

6、Response to comment:Mechanical properties are not reported for cladded thickness.

Response:We have added the hardness distribution and change analysis before and after the modification in row549 row to 570row .

7、Response to comment:I strongly advise authors to provide ethe detailed study of experimnetal and simulation part.

Response:Thanks to your hints, we have studied the spatial distribution of temperature characteristics of the electron beam cladding process, deeply simulated the solidification characteristics of the melt pool edge, and predicted the solidification organization. Also, we investigated the tissue distribution and the surface energy spectrum of the metallurgical bonding zone after the melting process, and studied the hardness of the cross-section before and after the melting process.

Round 2

Reviewer 2 Report

Accepted, the manuscript improved a lot.